# Unpacking Pandora from Its Box: Deciphering the Molecular Basis of the SARS-CoV-2 Coronavirus

**DOI:** 10.3390/ijms22010386

**Published:** 2020-12-31

**Authors:** Valerie Bríd O’Leary, Oliver James Dolly, Cyril Höschl, Marie Černa, Saak Victor Ovsepian

**Affiliations:** 1Department of Medical Genetics, Third Faculty of Medicine, Charles University, Ruska 87, Vinohrady, 10000 Prague, Czech Republic; marie.cerna@lf3.cuni.cz; 2Department of Experimental Neurobiology, National Institute of Mental Health, Research Programme 1, Topolova 748, 25067 Klecany, Czech Republic; cyril.hoschl@nudz.cz (C.H.); saak.ovsepian@nudz.cz (S.V.O.); 3International Centre for Neurotherapeutics, Dublin City University, Collins Avenue, Dublin 9, Ireland; oliver.dolly@dcu.ie; 4Department of Psychiatry and Medical Psychology, Third Faculty of Medicine, Charles University, Ruska 87, Vinohrady, 10000 Prague, Czech Republic

**Keywords:** COVID-19, virus, pandemic, bats, RNA, coronavirus, 2019-nCoV

## Abstract

An enigmatic localized pneumonia escalated into a worldwide COVID-19 pandemic from Severe Acute Respiratory Syndrome Coronavirus 2 (SARS-CoV-2). This review aims to consolidate the extensive biological minutiae of SARS-CoV-2 which requires decipherment. Having one of the largest RNA viral genomes, the single strand contains the genes *ORF1ab, S, E, M, N* and ten open reading frames. Highlighting unique features such as stem-loop formation, slippery frameshifting sequences and ribosomal mimicry, SARS-CoV-2 represents a formidable cellular invader. Hijacking the hosts translational engine, it produces two polyprotein repositories (pp1a and pp1ab), armed with self-cleavage capacity for production of sixteen non-structural proteins. Novel glycosylation sites on the spike trimer reveal unique SARS-CoV-2 features for shielding and cellular internalization. Affording complexity for superior fitness and camouflage, SARS-CoV-2 challenges diagnosis and vaccine vigilance. This review serves the scientific community seeking in-depth molecular details when designing drugs to curb transmission of this biological armament.

## 1. Introduction

Hindsight is 20/20, paraphrases knowledge of the correct way to proceed following an event. Five years ago, the American software developer Bill Gates warned in a Ted talk that if anything would kill millions of people in the next few decades, it would most likely be a highly contagious virus rather than a war [1]. In December 2019, a report emerged from the Wuhan Municipal Health Commission describing three patients with pneumonia of unknown etiology that was epidemiologically linked to an animal wholesale market in Hubei Province, China [2]. The outcome of the subsequent rapid response investigation by the Chinese Centre for Disease Control and Prevention revealed the causative agent of this enigmatic pneumonia to be a novel coronavirus named upon isolation as 2019-nCoV [3] and by the International Committee on Taxonomy of Viruses (ICTV) as SARS-CoV-2 (i.e., Severe Acute Respiratory Syndrome Coronavirus 2) [4]. The World Health Organization characterized the associated infection as COVID-19 (i.e., Coronavirus Disease 2019) [5]. Subsequent to the initial report, the COVID-19 epidemic in China became the first pandemic sparked by a coronavirus (WHO statement 11 March 2020) and to date has spread to every continent in the world except Antarctica. While the majority of the current literature on this topic focuses on the broad epidemiological implications of this ongoing pandemic, the purpose of this review is to overview the SARS-CoV-2 virus from a molecular perspective with implications for treatment to combat this novel virion.

## 2. Phylogenetic Profiling of the Novel Coronavirus

The initial COVID-19 patients in Wuhan provided bronchoalveolar-lavage fluid from which RNA was extracted, cloned, and subjected to third generation analysis using Illumina technology and Nanopore meta-transcriptomic sequencing [3]. Approximately 30,000 nucleotides were obtained within 20,000 viral readouts from the specimens and submitted to the National Center for Biotechnology Information (17 January 2020). Upon assembly, most contigs matched the genome of the subgenus *Sarbecovirus* (89.1% nucleotide similarity), genus *Betacoronavirus*, family *Coronaviridae* of the order *Nidovirales* [5]. Of note, however, despite the comparatively large sequence identity with other coronaviruses e.g., 87.9% sequence identity with bat-SL-CoVZC45 [6] and 79.5% sequence identity to SARS-CoV [7], SARS-CoV-2 had distinct gene sequences that enabled it to be regarded as a unique group 2B coronavirus [3]. The sequenced virus “Wuhan seafood market pneumonia virus isolates Wuhan-Hu-1” of 29,875 bp ss-RNA with the accession number MN908947 was subsequently renamed as “Severe acute respiratory syndrome coronavirus 2 isolate Wuhan-Hu-1” of sequence 29,903 bp with accession number NC_045512 (version 2), within the NCBI database [8]. This genomic sequence is one of the largest for all previously reported RNA viruses [9]. As all known human coronaviruses have animal origins, SARS-CoV-2 being classified as a member of the *Betacoronavirus* family was considered to have originated in a mammal most likely within a bat population. It has been suggested that SARS-CoV-2 probably emerged by genetic drift from bat-SL-CoV-RaTG13 [8]. The identification of an intermediate host has not yielded a definitive source to date.

## 3. The SARS-CoV-2 Genome Throws Up Unique Features in Stem-Loops

In common with other coronaviruses, the SARS-CoV-2 complete genome (NCBI reference sequence NC_045512, 29,903 bp) is composed of a single stranded RNA with base sequence orientation in a 5′ to 3′ direction typical of positive polarity and reflective of the eventual mRNA. The SARS-CoV-2 genome contains a 5′untranslated region (UTR; 265 bp), *ORF1ab* (21,289 bp) holding two overlapping open reading frames (13,217 bp and 21,289 bp, respectively) that encode two polyproteins (polyprotein 1a (pp1a) and polyprotein 1ab (pp1ab)), *S* (3821 bp), *ORF3a* (827 bp), *E* (227 bp), *M* (668 bp), *ORF6* (185 bp), *ORF7a* (365 bp), *ORF7b* (131 bp (3 bp overlap with *ORF7a*)), *ORF8* (365 bp), *N* (1259 bp), *ORF10* (116 bp), 3′ UTR (length 228 bp), and a 3′ poly A+ tail (length 33 bp) (Figure 1). Eight short nucleotide sequences (ranging in length from 6 to 49 bp) exist between each gene downstream of *ORF1ab*. It has been suggested that they represent short motifs called transcription-regulatory sequences (TRSs) playing a potential role in RNA polymerase jumping which assist in the production of sub-genomic RNAs (sgRNAs) [10]. To date, the function of these SARS-CoV-2 sgRNAs is unclear with speculation stemming from the production of such partial sequences for interference activity in other coronaviruses [11]. Like SARS-CoV [12], the RNA genome of SARS-CoV-2 probably has a 5′methylated cap due to the action of 2′-O-ribose methyltransferase (Nsp16) that it encodes. The translation of two polyproteins from a single mRNA occurs by ribosomal reading frameshifting within the specific RNA region 13,476–13,503 bp known as the “coronavirus frameshifting stimulation element stem-loop 1” [13] (Figure 2). This process has been described mainly in viruses (especially retroviruses [14]) with notable examples including human immunodeficiency virus for gag expression [15] and the influenza virus [16]. A tandem tRNA slippage mechanism enabling frameshifting in SARS-CoV has been characterized previously in cultured mammalian cells using mutagenic analysis, a dual luciferase reporter system and mass spectrometry [13]. This translational event is necessary for the synthesis of viral RNA polymerase and other enzymes (see below) which ensures that polyprotein 1a is expressed at specific levels relative to the products of the entire *ORF1ab* gene. Ribosomes shift translation frame at a slippery sequence U_UUA_AAC, known to be invariant among coronaviruses, after formation of a downstream RNA pseudoknot structure known as a “H-type” [13], an elaborated pseudoknot [17] or a “kissing stem loop” [18,19,20]. Clustal V alignment of SARS-CoV-2 and SARS-CoV revealed that the regions of the genome containing the frameshift site and that are potentially capable of “kissing” stem loop formation are identical between these virions except for an adenine in place of a cytosine at position 13,533 bp within *ORF1ab* for the novel coronavirus (Figure 2). Curiously, the sequence UUUAAAC is repeated eight times throughout *ORF1ab* (first base pair locations: 1664, 6085, 6750, 13,462, 16,669, 18,475, 20,227, and 20,817) and once within *S* (first base pair location: 24436); the implications for this are unknown. Like SARS-CoV, the novel coronavirus SARS-CoV-2 also contains a second overlapping potential shift site G_UUU_UUA (13,459–13,465 bp) where tandem slippage is hypothetically possible, but which has been ruled out for SARS-CoV given mutagenic analysis of in vitro expression constructs [13]. The proposed RNA secondary structure based on this alignment are shown (Figure 2). Both SARS-CoV and SARS-CoV-2 contain two sets of guanine: uracil base pairs, a unique feature at the base of the stem I pseudoknot with potential unconventional interaction. The importance of these unpaired nucleotides was demonstrated previously after mutagenic alteration of GU to AC in SARS-CoV reporter constructs reduced the ability of the pseudoknot to stimulate ribosomal frameshifting [13]. Upstream from the 3′UTR, a highly conserved RNA element known as the stem-loop II motif (s2m) has been identified in some coronavirus and astrovirus genomes [21] and has been well deciphered within SARS-CoV [12] (Figure 2A,B). Given the acquisition of a global folded tertiary structure comparable to that of the 530 loop of the 16S ribosomal RNA, it has been hypothesized that the function of the s2m is one of macromolecular mimicry to hijack the host translational machinery for use by the virus [12] (Figure 2A). The value of mimetic relationships instigates from the similarity between the signals emitted by distinct organisms which normally belong to different species [22] (Figure 2C,D). This would not be considered uncommon, as RNA viruses (e.g., turnip yellow mosaic virus) use biological mimicry as a key strategy for evading the immune system of the host [23] (Figure 2C). While SARS-CoV-2 contains the s2m element, it shows two variant nucleotides (cytosine at 29,732 bp and uracil at 29,758 bp) that differ from the SARS coronavirus (Figure 2). As this region is highly conserved in astrovirus, coronavirus and equine rhinovirus [12,21], any variation should be regarded as significant in the context of COVID-19, when designing drugs aimed at binding to the s2m element for tertiary structure disruption. This particularly applies to the uracil (29,758 bp; Figure 2 in red) whereby this alteration may change the interior of the molecule where Mg^2+^ potentially binds.

## 4. Polyproteins Translated From the SARS-CoV-2 Genome

The SARS-CoV-2 genome performs as a mRNA following cellular entry and is completely dependent on the translation machinery of the host cell. Ribosomal profiling has shown that translation of codons by rare tRNAs and non-cognate isoacceptor tRNAs (by wobble base pairing of codons and tRNAs) reduces translational efficiency [28]. Through the use of such techniques, coronavirus genomes including SARS-CoV-2 have been evaluated for the presence of so called “slow-codons” [29]. Results indicated that SARS-CoV-2 may have a higher protein translational rate compared to other coronavirus groups which have the ability to infect humans due to its low level of slow-codons [29]. Two long polyproteins are translated from SARS-CoV-2 mRNA which comprise the machinery that the virus needs for self-replication. These polyproteins include a replication/transcription complex, two proteases and structural proteins necessary for construction of new self-virions.

## 5. The Non-Structural Proteins Encoded by the SARS-CoV-2 Genome

The polyprotein 1a is proteolytically cleaved into eleven non-structural proteins (Nsp1–11). Polyprotein 1ab also contains these proteins along with five additional non-structural proteins (Nsp12–16) (Figure 3).

**Nsp1** (19.6 kDa; 180 amino acids) is known as the leader protein and colloquially referred to as the “cellular saboteur” [31] and “host shutoff factor” [32] due to an ability to divert proteins necessary for host translational mechanisms. It also prevents the host from assembling an antiviral arsenal [29]. The Nsp1 acts as a translation inhibitor via its C-terminal, blocking the ribosomal entry site to prevent host mRNA binding [32]. Of note, a SARS-CoV-2 genomic variant has recently been identified as a 9 bp deletion in position 686–694, corresponding to amino acids KSF in position 241–243, which alter Nsp1 interaction ability with implications for SARS-CoV-2 pathogenesis [33].

Selective pressure analysis of **Nsp2** (70.5 kDa, p65 homolog; 639 amino acids) noted a variation in SARS-CoV-2 that results in a glutamine at position 321 which has the potential to form H-bonds and may therefore confer increased stability over other sarcoviruses [34]. This alteration occurs in a domain that is homologous to an endosome-associated protein with a key role in avian infectious bronchitis virus pathology [34].

**Nsp3** (217 kDa; 1946 amino acids) produced by both pp1a and pp1ab, is the largest element of the replication and transcription complex (RTC) [35]. It contains multiple conserved domains that represent an N-terminal acidic phosphoesterase, a papain like protease (PLpro), Y-domain, transmembrane domain 1 (TM1) and an adenosine diphosphate-ribose 1′’-phosphatase (ADRP) (NCBI reference sequence YP_009725299) [2]. The papain-like protease (PLpro, PDB entry 4ow0) has a single subunit and uses a cysteine in a cleavage reaction. It is believed to make three specific cuts in the N-terminal of the polyprotein, and also to remove ubiquitin from ubiquitinated proteins in the host cell. Consequently, this process can interfere with the production of interferons necessary within the innate immune system leading to the short-circuiting of the host defense mechanism against SARS-CoV-2 [2]. Nsp3 interacts with the nucleocapsid protein (N), inhibitors of which might be useful for blocking SARS-CoV-2 replication [36].

**Nsp4** (56 kD; 501 amino acids) produced by both pp1a and pp1ab, includes a transmembrane domain (TM2). Based on a functional study of Nsp4 in murine hepatitis virus, it has been inferred that its glycosylation contributes to viral fitness [37]. Nsp4 is believed to complex with Nsp3 and Nsp6 for vesicle assembly within which viral replication occurs [38].

**Nsp5** (33.7 kDa, 3CLpro, Mpro, 307 amino acids) is the main proteinase [39]. On the basis of SARS-CoV research, it is presumed that Nsp5 in SARS-CoV-2 also mediates cleavage at eleven distinct sites to release Nsp4 to Nsp16 within pp1a and pp1ab, respectively [40,41,42]. Such cleavage also includes its own auto-proteolysis [40,41,42]. The proteases play essential roles in cutting the polyproteins into all of the functional units. The main protease (306 amino acid length, protein databank 6LU7 or Mpro [29]) is a dimer of two identical subunits that together form two active sites. Using the SARS-CoV protease as a reference, it is believed that the main protease of SARS-CoV-2 cuts the polyprotein at eleven sites.

**Nsp6** (33 kDa, 291 amino acids) has been shown from structural analysis to contain seven putative trans-membrane helices similar to other coronaviruses [43]. This protein locates to the endoplasmic reticulum (ER) and generates autophagosomes which are responsible for the delivery of cytoplasmic contents to lysosomes [34]. Multiple phenylalanine residues within Nsp6 are believed to favor more stable binding to the ER membrane compromising lysosomal delivery of coronaviral components destined for degradation [34]. Interestingly, amino acid change stability analysis of worldwide SARS-CoV-2 sequences identified a leucine37phenylalanine mutation that might contribute to lower stability of Nsp6 in some populations in Asia, America, Oceania, and Europe [34]. It has been speculated that this mutation may contribute to altered SARS-CoV-2 expression, influence host anti-viral defenses and significantly modify COVID-19 pathogenicity [34].

Conflicting evidence from SARS-CoV research suggests that **Nsp7** (9.2 kDa; 84 amino acids) and **Nsp8** (21.8 kDa; 199 amino acids) form either a hexadecamer (8:8) [44] or hetero tetramer (2:2) [45]. **Nsp8** forms a scaffold with head-to-tail interaction in which **Nsp7** subunits sandwich the scaffold without self-interaction. Of note, the NSP7-NSP8 complex acts as a primase for Nsp12, the RNA dependent RNA polymerase (RdRp) during viral replication [46].

**Nsp9** (12.3 kDa, 114 amino acids) from SARS-CoV-2 shares 97% sequence identity with that of SARS-CoV [47]. Based on this, it can be inferred that Nsp9 binds with low micromolar affinity to single stranded RNA [48] and most likely forms a complex with other Nsps for its role in viral replication [49]. Experiments with SARS-CoV suggest Nsp9 forms a dimer via an interaction motif GXXG, mutations of which inhibited efficient virus replication in vitro [50]. Recently, Nsp9 from SARS-CoV-2 has been purified as an obligate dimer and the crystal structure has been determined [47]. The structure of the SARS-CoV-2 Nsp9 showed conservation of a unique topological fold and the specific helical GxxxG dimerization interface when compared with homologues from SARS-CoV [47,48,49]. Such studies of the novel coronavirus Nsp9 will assist in drug screening strategies targeting the dimer interface with the view to compromising its replication ability. Recently, it has been reported that Conivaptan, Telmisartan, and Phaitanthrin D exhibited favorable docking scores against the target site of the Nsp9 replicase (PDB ID-6W4B) and suggests their potential usage as therapeutic agents against SARS-CoV-2 [51].

**Nsp10** (14.7 kDa; 140 amino acids; formerly known as a growth-factor like protein) stimulates the methyltransferase activities of **Nsp16** (2′-O-ribose methyltransferase) which leads to the modification of the cap structure present at the 5′ end of the SARS-CoV mRNA [52,53,54]. This camouflages the viral RNA from the host innate immune system and degradation by 5′-3′exoribonucleases [54]. Recently, potential inhibitors with capability to bind to the Nsp10/Nsp16 complex have been identified [55]. A secondary functional role can be predicted for Nsp10 in SARS-CoV-2 from data showing that it also interacts with **Nsp14**, a 3′-5′ exoribonuclease, in SARS-CoV [56].

**Nsp11** (1.3 kDa; 13 amino acids) is only produced by pp1a [34]. It is 92.3% identical between SARS-CoV-2 and SARS-CoV [57]. The function of this small peptide is unknown. **Nsp12** (106.6 kDa; 932 amino acids) is only produced by pp1ab and represents the RNA dependent RNA polymerase (RdRp) of SARS-CoV-2. It is an essential component of the replication-transcription complex of coronaviruses [40]. SARS-CoV-2 polymerase complex consists of the Nsp12 catalytic subunit and Nsp7-Nsp8 cofactors [24]. The polymerase domain adopts a structure resembling a cupped “hand” in resemblance with other polymerases [58]. Nsp12 polymerase is comprised of a ‘finger’ (398–581 and 628–687 a.a), a ‘palm’ (582–627 and 688–815 a.a), and a “thumb” subdomain (816–919 a.a). Nsp12 is 96% and 71% identical to that of SARS-CoV and MERS-CoV respectively [57], with the majority of sequence variation present in the N-terminal region [59]. Despite this sequence diversity, the RdRp motifs (A–G) are highly conserved among the three coronaviruses [57]. Nevertheless, SARS-CoV-2 has been shown using multiple sequence alignment to hold three substitutions compared to SARS-CoV (Motif A: T614N; Motif C: Y769F; Motif D: A787S) [59]. It has been proposed that RdRp motifs (A–G) constitute the polymerase active site, with an open nucleotide triphosphate (NTP) entry tunnel that leads to the catalytic center [60]. Except for Motifs D and G, all others directly take part in NTP binding/hydrolysis. Motifs A and C hold catalytic site carboxylates and motif B binds the NTP base/sugar moiety. Motif E referred to as the “primer grip” in SARS-CoV [61] is near the NTP binding region. Motif F interacts with the triphosphate moiety of NTP [59]. In depth interaction analysis revealed functionally important aspartate residues in Motif A (Asp623) and C (Asp760), along with conserved Arginine residues within Motif F (Arg553 and Arg555), and C (Ser759) which interacted strongly with cytochrome inhibitor compounds (CMP2, CMP17a and CMP21) [58]. The most promising anti-SARS-CoV-2 drugs are RdRp inhibitors (e.g., Remdesivir) which represent nucleos(t)ide analogues (NAs). Upon delivery into the host cell, nucleoside/nucleotide prodrugs are metabolized into an active 5′triphosphate form (5′-TP) which compete with endogenous nucleotides as substrates for the SARS-CoV-2 RdRp to be incorporated into the nascent RNA and elicit an antiviral effect. **Nsp13** (66.8 kDa; 601 amino acids) is produced by the pp1ab only and serves as a helicase to unwind SARS-CoV-2 RNA making it accessible to other interacting factors. The general structure of SARS-CoV-2 Nsp13 is believed to be a triangular pyramid shape made up of five domains, zinc binding domain (ZBD), stalk domain (S), RecA-like domains (1A, 2A: for ATP binding and hydrolyzing nucleotides) and 1B (forming the base of the triangle structure) [58]. The SARS-CoV-2 Nsp13 has similar conserved NTPase active site residues located within the cleft between domain 1A and 1B (including Lys288, Ser289, Asp374, Glu375, Gln404, and Arg567) as present in SARS-CoV [58]. Small molecules able to inhibit the NTPase activity by interferences with ATP binding have been proposed as an ideal strategy to develop SARS-CoV-2 inhibitors [58].

**Nsp14** (59.8 kDa; 527 amino acids) is a 3′ to 5′ exoribonuclease within the pp1ab polypeptide. It was shown that Nsp14 is activated by Nsp10 and associated cofactors (Nsp7 and Nsp8) leading to increased viral proof-reading capacity [54,60]. Anti-SARS-CoV-2 nucleoside analogue design must consider a faster rate of incorporation by the trimeric RNA polymerase complex (Nsp12, Nsp7, and Nsp8) than that of excision by N-terminal domain of Nsp14 [54,60]. Coupling nucleoside analogues with exonuclease inhibitors may be a strategy worth considering to reduce SARS-CoV-2 therapeutic resistance [60].

**Nsp15** (38.8 kDa; 346 amino acids) is an endoRNase produced by pp1ab that serves as a degrader of viral RNA fragments that may activate the infected cell’s antiviral defenses. The crystal structure of SARS-CoV-2 Nsp15 has been recently deciphered and shown to be very similar to that of SARS-CoV and MERS-CoV homologues [62]. The catalytic function of Nsp15 resides in the C-terminal NendoU (Nidoviral Uridylate-specific Endoribonuclease) domain. The active site, located in a shallow groove between the two β-sheets, carries six key residues (His235, His250, Lys290, Thr341, Tyr343, and Ser294) which are conserved amongst SARSCoV-2, SARS-CoV, and MERS-CoV proteins [62]. Based on the structural comparisons, it has been suggested that inhibitors of SARS-CoV Nsp15 may also inhibit the SARS-CoV-2 homolog [62]. Computational molecular docking simulations of a range of Chinese traditional medicine Saikosaponins, noted that Saikosaponin V has high affinity binding to the narrow binding pocket of NSP15 [63]. It has been suggested that Saikosaponin V may represent an inhibitor targeting SARS-CoV-2 via Nsp15 interaction [63].

**Nsp16** (33.3 kDa; 298 amino acids) is a 2′-O-ribose methyltransferase produced by pp1ab. It is assumed that Nsp16 in SARS-CoV-2 is activated by Nsp10 [64]. It is believed that Nsp16 plays an essential role in coronavirus mRNA cap 2′-O-ribose methylation and that the presence of N7-methyl guanosine is a prerequisite for Nsp16 binding [64]. The purpose of this activity is one of camouflage as a means of evading detection by the hosts immune system.

**Accessory proteins (ORF3a, 6, 7a, 7b, 8, 9b, 9c, and 10).** The SARS-CoV-2 genome encodes several unidentified non-structural open-reading frames [3]. Some of these open reading frames are translated into accessory proteins ORF3a, 6, 7a, 7b, 8, and 10. ORF3b might not be translated.

The SARS-CoV-2 **ORF3a** protein has six functional domains (I to VI), three trans endoplasmic reticulum (ER) membrane regions [65] and holds 72% sequence similarity to that detected in SARS-CoV [62]. Interestingly, micro-clonality has been observed in ORF3a of SARS-CoV-2 due to non-synonymous mutations causing the isolates to cluster into defined phylogenetic clades representing distinct subpopulations [65]. Domain III which consists of a K+ channel in SARS-CoV was found to hold several mutations within this domain in SARS-CoV-2 [65]. Of importance is H93Y, as this mutation has previously been linked in SARS-CoV to the loss of the K+ channel and reduced pro-apoptotic activity [9].

The function of **ORF6** in SARS-CoV-2 can be speculated from studies of this protein in SARS-CoV. It was demonstrated that the SARS-COV ORF6 protein localized to the ER/Golgi membrane in infected cells, where it bound to and disrupted nuclear import complex formation via tethering karyopherin alpha 2 and karyopherin beta 1 to the membrane [66]. It was believed that retention of these import factors at the ER/Golgi membrane lead to a loss of STAT1 transport into the nucleus in response to interferon signalling [66]. In this way the virion blocked the expression of STAT1-activated genes involved in the antiviral innate immune response.

**ORF7a** has been nick-named “the virus liberator”, as it has the ability in SARS-CoV to break a viral antagonist BST-2/Tetherin [67]. A deletion of 27 amino acids which maps to a putative signal peptide within ORF7a has recently been reported in a SARS-CoV-2 sample taken in Arizona USA; the implications for viral fitness and the prevalence of this 81 bp mutation are currently unknown [68].

**ORF7b** overlaps ORF7a and has been studied in SARS-CoV using in vitro translation mechanisms [69]. ORF7b is a highly hydrophobic protein but, so far, its function in SARS-CoV and SARS-CoV-2 remains unknown.

**ORF8** of SARS-CoV-2 does not contain any functional domain/motif [54] or a 29-nucleotide deletion which is found in some strains of SARS-CoV resulting in the formation of ORF8a and ORF8b [70]. Two missense mutations (28077G->C, 28144T->C) have been found in ORF8 of SARS-CoV-2. These have resulted in amino acid changes (V62L and L84S) within this accessory protein [71]. It is speculated that ORF8 in SARS-CoV-2 may encode a secreted protein with an alpha-helix and a six-stranded beta sheet [34]. An aggregation motif VLVVL (amino acid 75–79) which has been found in SARS-CoV ORF8b and shown to trigger intracellular stress pathways and activate inflammasomes [72], is absent in ORF8 of SARS-CoV-2 [34].

**ORF9b** and **ORF9c** are encoded by a region of the genome that overlaps the gene for the nucleocapsid in SARS-CoV-2. Limited information is available for these accessory proteins. A non-peer reviewed proteomic interaction study suggests that ORF9b is a signalling molecule and ORF9c may have a mitochondrial role in SARS-CoV-2 [57]. Through the use of SARS-CoV-2 baits, interaction of ORF9c with the Respiratory complex 1 was noted [57]. Drug-target associations from chemoinformatic searches found amongst others, Midostaurin (protein kinase inhibitor) and Metformin (mannose receptor 1 inhibitor) interacted with ORF9b and ORF9c respectively [57].

Unlike SARS-CoV, **ORF10** (38 amino acids in length) exists in SARS-CoV-2 and may encode a functional transmembrane protein under positive selection pressure [71]. ORF10 is also potentially encoded by pangolin (RaTG13, Gd/1 and Gx/P1E) and bat (SL-CoV2C45) viruses [71]. However, it is unlikely to be expressed in SARS-CoV-2, with calls for the annotation of ORF10 to be reconsidered [10].

## 6. SARS-CoV-2 Structural Proteins

SARS-CoV-2 is an enveloped virus, approximately 120 nm in diameter [73] and similar to other coronaviruses with a lipid membrane derived from the host cell that serves as an embedding edifice for surface proteins [74]. In total SARS-CoV-2 has four major structural proteins: (1) spike, (2) envelope, (3) membrane, and (4) nucleocapsid. This novel coronavirus derives its name from the glycoprotein trimer spike that protrudes from the lipid membrane [31]. It is believed that the spike is responsible for the cellular internalization of SARS-CoV-2, host tissue tropism and coronavirus transmission capacity [75]. It has been suggested that the SARS-CoV-2 Spike differs by 12.8% from that of SARS-CoV [51]. While such variation in the spike protein amino acid composition exists, this study reported no overall difference in their structures. Of note however, was the discovery of various novel N- and O linked glycosylation sites (e.g., NGTK, NFTI, NLTT, and NTSN) in the spike of SARS-CoV-2 compared to SARS-CoV [51,76]. It has been proposed that such unique features contribute to the shielding and camouflage of SARS-CoV-2 from the defense system of the host [76]. Interestingly, a potential interaction between the S1 domain of the spike glycoprotein and human CD26 may occur, highlighting a probable mechanism enabling SARS-CoV-2 to hijack human cells [76].

The spike glycoprotein trimer has three receptor binding domains (RBDs) which are responsible for SARS-CoV-2 entry into host cells via ACE2 (i.e., Angiotensin Converting Enzyme 2). In the predominant state of the trimer one of the three RBDs is rotated upwards in an ACE2 accessible conformation [31]. The RBD amino acid sequence of SARS-CoV-2 and SARS-CoV are 72% identical with a very similar ternary structure [29]. A comparison of S between SARS-CoV-2 and closely related SARS-CoV-like viruses revealed that the majority of residues important for ACE2 engagement are not conserved in the novel coronavirus [77]. SARS-CoV-2 has a distinct flexible loop in the RBD due to glycyl replacing rigid prolyl residues [29]. A unique phenylalanine F486 in this loop may enable the penetration of SARS-CoV-2 into a deep hydrophobic pocket in ACE2 [29], perhaps contributing to its higher affinity of 10–20% compared to SARS-CoV for this receptor [31]. Of note, is the recent discovery of a furin cleavage site (681-PRRA-684) unique to SARS-CoV-2, which is speculated to provide a gain-of-function for this novel virion enabling more efficient infection in the human host compared to other lineage *betacoronaviruses* [78]. Dynamic tracking of S amino acid variants has been underway throughout 2020. By April 2020, the D614G mutation in S was evident in Europe and Africa, becoming the globally dominant form of the virus within months [79]. Other S variants (N501Y and H69/V70) have appeared with evidence of associated increased transmission. As of December 2020, the SARS-CoV-2 B.1.1.7 variant which originated in Southeast England (United Kingdom) represents 23 separate mutations, eight of which are in S, boosting transmission rates by seventy percent [80].

The **lipid envelope** (E) of SARS-CoV-2 is encoded by an evolutionary conserved region of its genome, having a sequence identity of 94–96% with SARS-CoV [7]. It is a small transmembrane non-glycosylated, homo-pentameric protein according to template searching results using Swiss Model analysis [29]. The envelope gene shows codon usage bias and is highly expressed given its reported low ENc (i.e., effective codon usage) values [81].

The **membrane** (M) protein of SARS-CoV-2 like other coronavirus may be responsible for virion assembly and might comprise ion channel actions [82]. Predicted intrinsic disorder (PID) values for the membrane protein have been evaluated based on the premise that coronaviruses that remain in harsh environments require harder shells to survive (i.e., are less disordered) [83]. The membrane protein of SARS-CoV-2 has been reported to be amongst the hardest in the coronavirus family, with a PID model suggesting that SARS-CoV-2 is therefore more likely to have greater resilience in body fluids and the environment than SARS-CoV and MERS-CoV [83].

The **nucleocapsid** (N) protein is the most abundant protein produced by the SARS-CoV-2 genome [84]. It is a protein that is reportedly well conserved across coronaviruses [85]. However, it has been noted that the higher PID levels within the SARS-CoV-2 nucleocapsid compared to other coronaviruses, might contribute to its greater infectivity and higher respiratory transmission potential [83]. The primary role of the N protein is to package the viral genome into long, flexible, helical ribonucleoprotein (RNP) complexes [86]. Weak specific interaction between the single stranded RNA and N proteins relies on packaging signals of 2–4 bps throughout the SARS-CoV-2 genome [87]. It is believed that longer motifs for packaging would be too restrictive due to the protein coding function of the SARS-CoV-2 genome [87]. Nevertheless, a long packaging motif (UAUUCAAACAAUUGUUG) has been identified in isolates of this novel coronavirus [87]. Considering the variable nature of the spike protein, N should also be considered a more promising therapeutic target as it is more stable against point mutation load [87]. The SARS-CoV-2 nucleocapsid protein shares approximately 90% amino acid identity with that of SARS-CoV [84]. Nevertheless, N antibodies against SARS-CoV fail to provide immunity against SARS-CoV-2 infection [84]. Further comparative multiple sequence alignment of the N proteins from SARS-CoV-2 and SARS-CoV revealed the presence of a novel large insert (25–41 residues depending on the alignment method) in the former between two putative functional domains [88]. These domains have been reported to be homologous to the N-terminal domain and adenosine diphosphate ribose 1 phosphatase (or Macrodomain 1 [89]) of SARS-CoV (86). This large peptide insert specific for SARS-CoV-2, tentatively matched (sequence identity 46%) that of C-Jun-amino-terminal kinase-interacting protein 4 of a species of fish *Labrus bergylta* [88].

## 7. Future Mitigation Plans

The coordination of public health bodies and research institutions will be vital for the continuous evaluation of COVID-19 in populations worldwide. Such appraisals will involve vigorous assessment of the effectiveness of targeted interventions such as lockdown or individualized quarantine along with improved diagnostic tests. Rapid COVID-19 vaccine development has occurred at an unprecedented level, giving rise to global procurement mechanisms for fair distribution. COVID-19 will be viewed as a strong justification for universal healthcare which will include the benefits of collective diagnostic testing approaches. The ability to use COVID-19 status or previous exposure (i.e., acquired immunity) will influence hospital isolation practices, neonatal care and guide the use of personal protective equipment. Such measures should become the normal standard of patient care, safeguarding societies to avert recurrent widespread COVID-19 circulation. Sensible social distancing measures and heightened environmental hygiene standards, will contain the SARS-CoV-2 pandemic.

This overview will support the scientific community seeking in depth molecular details when designing drugs to curb this biological armament or prevent another variant coronavirus transmitting to our species.

## Figures and Tables

**Figure 1 ijms-22-00386-f001:**
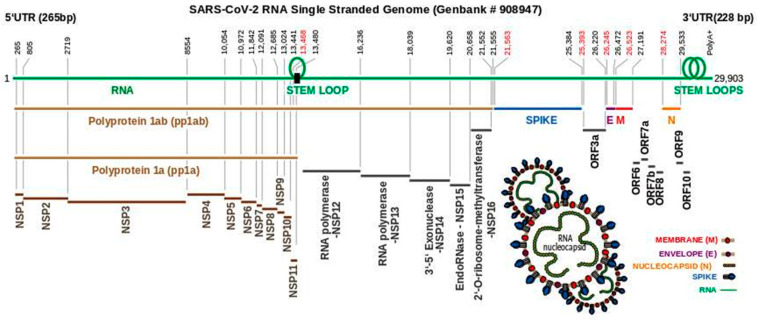
Schematic overview of the SARS-CoV-2 single stranded RNA genome. Upper line (green) represents the Severe Acute Respiratory Syndrome Coronavirus 2 isolate Wuhan-Hu-1 accession number NC_045512 version 2 entry in the National Centre for Biotechnology Information database. Numbers (black) represent the beginning and end of a coding region. Of note, coding regions do not run consecutively (numbers in red). Polyproteins 1ab (pp1ab) and 1a (pp1a) are translated from *ORF1ab* (265–21,555 bp) via a ribosomal slippage site (black rectangle) and stem loop formation. Non-structural proteins (NSP 1–11) are cleaved from both polyproteins. Cleavage of NSP12–16 occurs only from pp1ab enabling the formation of various listed enzymes. Structural proteins spike (S), envelope (E), membrane (M) and nucleocapsid (N) are encoded by the genes *S* (21,563–25,384 bp), *E* (26,245–26,472 bp), *M* (26,523–27,191 bp), and N (28,274–29,533 bp). Open reading frames (ORF) ORF3a, 6, 7a, 7b, 8, 9, and 10 are encoded by intermediate genomic segments as indicated. The location of two stem loops is shown adjacent to the 3′UTR representing pseudoknot stem loop (left) and s2m (right). The overall structure of SARS-CoV-2 is illustrated with abbreviations as indicated above.

**Figure 2 ijms-22-00386-f002:**
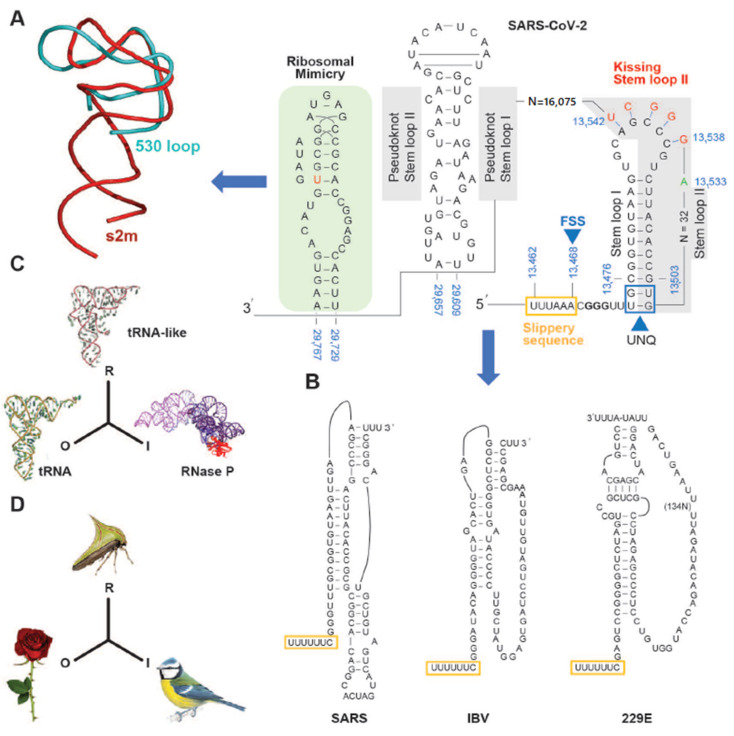
Sequences within the SARS-CoV-2 genome are involved in ribosomal frameshifting and RNA stem-loop structural formation. (**A**) The predicted standard coronavirus heptanucleotide shift site U_UUA_AAC is present and shown with the slippery sequence (brown box) and frame shift site (FSS) followed by a guanine conservation sequence (bold). An unpaired G:U nucleotide quartet (UNQ: boxed) may offer stability to the stem loop I and II. Binding between the stem loops and a downstream nucleotide region (13,538–13,542 bp) results in “kissing” stem loop II formation. A variant in SARS-CoV-2 (13533C->A) compared to SARS-CoV is indicated. Potential pseudoknot stem loops occur between 29,609–29,657 bp (grey box). Conservation element s2m (stem-loop II motif, pale green box) is shown that may partake in 16S ribosomal RNA macromolecular mimicry. Ribbon diagram (left of arrow) showing the similarity in the backbone folds between the s2m structure (red) and the 530 loop of the 16S rRNA (blue). (**B**) Sequence and base-pairing of three different coronavirus stimulatory elements; SARS three-stemmed pseudoknot [24,25], the avian infectious bronchitis virus (IBV) two-stemmed pseudoknot [26], and the kissing loops of the human coronavirus 229E stimulatory element [17]. The heptameric slippery sites are shown (yellow rectangles). (**C**,**D**) Models of mimetic relationships corresponding to the representamen: R, the model to the object: O and the operator to the interpretant: I [27]. A molecular example (**C**): the tRNAPhe is considered to be the object, the untranslated end of the plant virus TYMV genome is the representamen, while the interpretant is the tRNA precursor processing enzyme RNase P Modified from [23]. In (**D**), the plant’s thorn is the object, the thorn bug is the representamen and the predatory bird that the prey eludes is the interpretant. Modified from [23].

**Figure 3 ijms-22-00386-f003:**
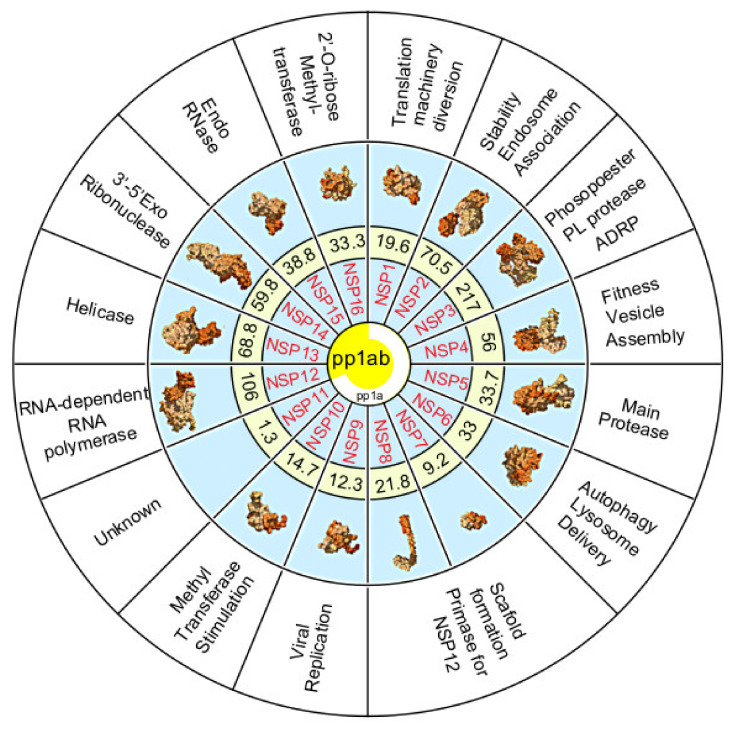
Summary diagram illustrating the non-structural proteins (Nsp) 1–16 cleaved from polyproteins pp1a (white partial circle) or pp1ab (yellow center). Numbers represent the molecular weight of Nsps in kilodaltons. Secondary structure prediction of each Nsp is shown except for Nsp11 which is unknown (Source PDB101.rcsb.org [30]. Nsp function is provided in the outer ring.

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
