# Peer review of "Unpacking Pandora from Its Box: Deciphering the Molecular Basis of the SARS-CoV-2 Coronavirus"

_ijms, 2020, doi:10.3390/ijms22010386_

Round 1

Reviewer 1 Report

The manuscript by O’Leary et al presents a summary of our basic understanding of SARS-CoV-2 till May 2020. In their abstract, the authors promise “extensive biological minutiae” and “in-depth molecular details”. This seems a worthy endeavour given our ever-expanding information on SARS-CoV-2. Unfortunately, however, the manuscript is a little hard to get through, only scratches the surface of the molecular principles underlying SARS-CoV-2 infections, and is not very up to date. A lot of new information has been published recently and this is not covered. While it is understandable that it is hard to keep track of and keep up with all developments, in its present form this manuscript does not add any new insights or depth to the current literature.

Main points:

  1. What is the innovative aspect of this review compared to recent and more in-depth reviews on this topic literature (e.g. Hartenian et al, J Biol Chem 2020)?
  2. Other than frame-shifting, few molecular details about the viral genome are discussed. How does the virus transcribe its genome (and express the downstream ORFs, make subgenomic RNAs, etc)? How does it replicate? How is the viral genome packaged?
  3. No information is given about host protein or membrane use, which are both important in the viral infection cycle.
  4. Details on the viral proteases and their function during infection (cleavage of viral and host prpoteins) is limited.
  5. Details on nsps and spike are not up to date.
  6. Figures 2B, C and D are not referenced in the text or discussed.
  7. The text is hard to get through. This is in part caused by the word choice and poor grammar, but also by the formatting (e.g. part about nsps is mostly one block of text that is not consistently formatted).
  8. It might be beneficial to include a figure to represent the phylogenetic profiling (part 2) and compare seasonal and pandemic coronaviruses.

Minor points:

  1. Line 15. Can a pneumonia really be a pandemic or is this a property of the virus?
  2. Line 15: The term “enigmatic” is a bit hyperbole and does not reflect the extensive amount of knowledge acquired before and during the pandemic.
  3. Line 17. Why does “consolidate … minutiae” require decipherment? This sentence currently suggests that the author’s work needs decipherment, but this is probably not what they mean. Please correct grammar.
  4. Line 18. The nucleoprotein is abbreviated as N in the text. In the abstract it is NC. See also line 123-124.
  5. Line 18. Why do the authors distinguish between genes and open reading frames?
  6. Line 24: Why does SARS-CoV2 have superior fitness and how?
  7. Line 67. Can a virus genome have a locus?
  8. Line 74. Where do the authors get the information that the polyA tail is only 33 nt long? Is this based from the consensus sequence file? Experimental evidence, e.g. Spagnolo et al 2001, suggests that coronavirus polyA tails are actually longer than 50 nt
  9. The title does not reflect the content of the review which does not sufficiently addresses the molecular basis of SARS-Cov-2.
  10. Line 94 “implications for this are unknown”. The presence of a sequence is not important per se. It has to occur in the context of an RNA structure and in the right translation frame to work. The authors should discuss this.
  11. Figure 2C/D. It is unclear why the authors are comparing the evolution of a thorn and pointy shapes, which have entirely different functions in the organisms shown in Fig. 2D (they just look alike), with the similar the similar shape and function of the t-RNA structure.
  12. Line 121: Adding proteases involved in the polyprotein proteolysis could be relevant.
  13. Line 179, Line 279 and figure 3: the mass of nsp16 is not 333 kDa.
  14. Line 122. Spike is not a human protein and does not need to be capitalised. It is also commonly abbreviated as “S".
  15. Line 110 “this region is highly conserved”. How can this sequence be highly conserved but variable in SARS-CoV-2? In which viruses is it highly conserved? Context needs to be provided.
  16. Line 89: clustal V does not exist. The authors probably mean W or Omega.
  17. Line 153: It would be appropriate to explain better why SARS-CoV-2 has a higher translation rate compared to other coronaviruses and why this is relevant.
  18. Line 166: The references lack relevance and up to date information.
  19. Line 364. Explain PID.

Author Response

Response to the reviewer:
The manuscript by O’Leary et al presents a summary of our basic understanding of SARS-CoV-2 till May 2020. In their abstract, the authors promise “extensive biological minutiae” and “in-depth molecular details”. This seems a worthy endeavour given our ever-expanding information on SARS-CoV-2. Unfortunately, however, the manuscript is a little hard to get through, only scratches the surface of the molecular principles underlying SARS-CoV-2 infections, and is not very up to date. A lot of new information has been published recently and this is not covered.
Response: The authors would like to highlight that this review is intended to provide details of the molecular basis of the virus and is not intended to explore the underlying principles of COVID-19 infection. The manuscript has been updated in various sections with recent findings and with data from early 2020 still providing a valuable information source for the scientific community.
While it is understandable that it is hard to keep track of and keep up with all developments, in its present form this manuscript does not add any new insights or depth to the current literature.
Response: The authors would strongly disagree. This review expands our recent Genome of the Month short overview article in Trends in Genetics which was voted amongst the most read articles since 2015 and listed in the top 10 WHO for essential reading. There has been requests for details to be provided in the literature in relation to molecular basis of the SARS-CoV-2 virus.
Main points:
1. What is the innovative aspect of this review compared to recent and more in-depth reviews on this topic literature (e.g. Hartenian et al, J Biol Chem 2020)?
Response: The recent paper by Hartenian et al JBC 2020, is indeed a very in-depth excellent review which presents the molecular virology of coronavirus infection (ie. it covers a much broader topic) including cellular entry, remodelling of the intracellular environment and the multifaceted immune evasion strategies. These aspects are not covered in our manuscript, which aims to focus on SARS-CoV-2 and the molecular basis of this novel virus. While Hartenian et al mention the ribosome frameshifting site in two sentences and illustrates it as a dot in figure 3, our manuscript provides in Figure 2 the precise sequences and secondary structural details involved in kissing loop formation, frameshifting and mimicry. The details are necessary for therapeutic design and will serve as a valuable information source for researchers tasked with the examination of extensive databases.
2. Other than frame-shifting, few molecular details about the viral genome are discussed.
Response: The manuscript presents molecular details as were available up until May 2020. The authors have updated the text to reflect current information that has arisen since submission. How does the virus transcribe its genome (and express the downstream ORFs, make subgenomic RNAs, etc)? How does it replicate? How is the viral genome packaged?
Response: The authors wish to thank the reviewer for these important points. Since writing this manuscript, Kim et al utilised innovative sequencing techniques to provide a high-resolution map of the SARS-CoV-2 transcriptome. Results showed its high complexity from discontinuous transcription which produces 9 subgenomic RNAs. The manuscript has now been updated to include this information and an explanation of short transcription-regulatory sequences (TRSs) in regards to the very short
intermediate sequences between ORFs. The non-structural proteins involved in replication are detailed extensively. The revised manuscript now includes the role of N and ssRNA motifs in viral packaging.
3. No information is given about host protein or membrane use, which are both important in the viral infection cycle.
4. Response: Please be aware that the topic of this review is not focused on viral infection.
5. Details on the viral proteases and their function during infection (cleavage of viral and host prpoteins) is limited.
Response: Please be aware that the topic of this review is not focused on viral infection.
6. Details on nsps and spike are not up to date.
Response: The manuscript has been updated for the non-structural proteins.
The text now includes further information related to the spike variants.
7. Figures 2B, C and D are not referenced in the text or discussed.
Response: This omission has been corrected in the revised manuscript with references provided.
8. The text is hard to get through. This is in part caused by the word choice and poor grammar, but also by the formatting (e.g. part about nsps is mostly one block of text that is not consistently formatted).
Response: The authors would like to remind the reviewer that this article was written by a native English speaker. The other two reviewers had no issue with the standard of language used in the text.
The formatting for the non-structural protein section of the manuscript has been corrected.
9. It might be beneficial to include a figure to represent the phylogenetic profiling (part 2) and compare seasonal and pandemic coronaviruses.
Response: Thank you for the suggestion. A phylogenetic profile has been published in our recent TIG article. Furthermore, a comparison of seasonal and pandemic coronaviruses is outside the remit of this review which focuses on just SARS-CoV-2.

Minor points:
1. Line 15. Can a pneumonia really be a pandemic or is this a property of the virus?
Response: The text reads the causative agent of this enigmatic pneumonia to be a novel coronavirus.
2. Line 15: The term “enigmatic” is a bit hyperbole and does not reflect the extensive amount of knowledge acquired before and during the pandemic.
Response: This word refers to its initial appearance in China when nothing was known about this virus.
3. Line 17. Why does “consolidate … minutiae” require decipherment? This sentence currently suggests that the author’s work needs decipherment, but this is probably not what they mean. Please correct grammar.
Response: The text in the manuscript states ‘the extensive biological minutiae of SARS-CoV-2 which requires decipherment’ – the authors do not see any indication that this sentence refers to the authors work.
4. Line 18. The nucleoprotein is abbreviated as N in the text. In the abstract it is NC. See also line 123-124.
Response: Thank you. The correction has been made.
5. Line 18. Why do the authors distinguish between genes and open reading frames?
Response: The open reading frames contain several genes.
6. Line 24: Why does SARS-CoV2 have superior fitness and how?
Response: This term is used in the abstract to broadly overview the comparative basis of the whole manuscript.
7. Line 67. Can a virus genome have a locus?
Response: This was the NCBI categorisation for the code NC_045512. The text now reads NCBI reference sequence instead of locus.
8. Line 74. Where do the authors get the information that the polyA tail is only 33 nt long? Is this based from the consensus sequence file? Experimental evidence, e.g. Spagnolo et al 2001, suggests that coronavirus polyA tails are actually longer than 50 nt.
Response: According to the reference sequence NC_045512 the poly A tail is between 29871 – 29903bp, ie. 33bps
9. The title does not reflect the content of the review which does not sufficiently addresses the molecular basis of SARS-Cov-2.
Response: The authors respectfully disagree.
10. Line 94 “implications for this are unknown”. The presence of a sequence is not important per se. It has to occur in the context of an RNA structure and in the right translation frame to work. The authors should discuss this.
Response: The authors would like to remind the reviewer that the sequence in question codes for a ‘slippery-sequence’ where potential ribosomal frameshifting may occur. The authors agree that the secondary structure would most likely be needed considering the kissing loops formation requirement of frameshifting. To date this is speculative and as such discussion might be premature.
11. Figure 2C/D. It is unclear why the authors are comparing the evolution of a thorn and pointy shapes, which have entirely different functions in the organisms shown in Fig. 2D (they just look alike), with the similar the similar shape and function of the t-RNA structure.
Response: The text has been updated to include more information about biological mimicry.
12. Line 121: Adding proteases involved in the polyprotein proteolysis could be relevant.
Response: The authors prefer to maintain the text as it is to avoid confusion, considering the relevant proteases are introduced later in the text along with their abbreviations.
13. Line 179, Line 279 and figure 3: the mass of nsp16 is not 333 kDa.
Response: Apologies, the molecular weight of nsp16 is 33324g per mole or 33.3 KDa.
14. Line 122. Spike is not a human protein and does not need to be capitalised. It is also commonly abbreviated as “S".
Response: The manuscript has been corrected in light of this comment.
15. Line 110 “this region is highly conserved”. How can this sequence be highly conserved but variable in SARS-CoV-2? In which viruses is it highly conserved? Context needs to be provided.
Response: The text now states that this region is highly conserved in astrovirus, coronavirus and equine rhinovirus with references provided.
16. Line 89: clustal V does not exist. The authors probably mean W or Omega.
Response: Clustal V does exist. Here is a link: https://doi.org/10.1093/bioinformatics/8.2.189
17. Line 153: It would be appropriate to explain better why SARS-CoV-2 has a higher translation rate compared to other coronaviruses and why this is relevant.
Response: The text now includes the words ‘due to its low level of slow-codons’. This is relevant due to the resulting higher translation rate as already stated in the text.
18. Line 166: The references lack relevance and up to date information.
Response: The text has been updated to include the data from Chandel et al 2020 Journal of biomolecular structure and dynamics.
19. Line 364. Explain PID.
Response: The abbreviation is provided in the text along with reference 83 where it is explained in detail.

Reviewer 2 Report

The work presented by O'Leary and colleagues on the molecular basis of the SARS-CoV-2 Coronavirus is an exhaustive review of the molecular components and mechanisms of action of SARS-CoV-2 that can support other researchers.

Minor revision concerns:

Line 1: the introductive sentence is not very clear. I suggest rephrasing.

Figure 2: I suggest to increase the quality of the image

Figure 3: remove the "Figure 3" on the left corner of the image

Conclusion paragarph: I would suggest to stress out more how this review can support the scientific community.

Author Response

Response to reviewer 2:

The work presented by O'Leary and colleagues on the molecular basis of the SARS-CoV-2 Coronavirus is an exhaustive review of the molecular components and mechanisms of action of SARS-CoV-2 that can support other researchers.

Response to reviewer: The authors wish to thank the reviewer for acknowledging the review as an extensive resource of use for the scientific community.

Minor revision concerns:

Line 1: the introductive sentence is not very clear. I suggest rephrasing.

Response: The first sentence of the introduction has been slightly changed for clarification purposes.

Figure 2: I suggest to increase the quality of the image.

Response: The figure is now submitted as an Adobe Illustrator file with increased image quality.

Figure 3: remove the "Figure 3" on the left corner of the image

Response: This has been removed from the left corner of the image.

Conclusion paragraph: I would suggest to stress out more how this review can support the scientific community.

Response: We appreciate the suggestion. The corrected manuscript now includes the following as a final sentence: ‘This overview will support the scientific community seeking in depth molecular details when designing drugs to curb this biological armament or prevent another variant coronavirus transmitting to our species’.

Reviewer 3 Report

In the present review the authors focused on the biological function of COVID-19, taking into account many scientific studies in order to decipher one of the largest RNA viral genomes, which are critical for the development of new therapeutic strategies for this novel coronavirus, waiting for an effective vaccination.

Overall, the work is well structured, exhaustive and the references are adequate.

The review includes a comprehensive view of the research area, but some figures  figures require improvement in sharpness (e.g. figure 1 and 2). 

Author Response

Response to Reviewer 3:

In the present review the authors focused on the biological function of COVID-19, taking into account many scientific studies in order to decipher one of the largest RNA viral genomes, which are critical for the development of new therapeutic strategies for this novel coronavirus, waiting for an effective vaccination.

Overall, the work is well structured, exhaustive and the references are adequate.

Response: The authors wish to thank the reviewer.

The review includes a comprehensive view of the research area, but some figures require improvement in sharpness (e.g. figure 1 and 2).

Response: The figures have now been uploaded as Adobe Illustrator original files in order to improve the resolution quality.

Round 2

Reviewer 1 Report

The manuscript is much improved and the work is now a helpful overview of the literature. The authors should still fix the error in Fig. 3, where nsp16 (2'O-ribose methyltransferase) is indicated to have a mass of 333 kDa. This should be 33 kDa.
